# Adherence to different complementary feeding methods in the first year of life: A randomized clinical trial

Christy Hannah Sanini Belin[1]*, Leandro Meirelles Nunes[1,2‡], Cátia Regina Ficagna[3‡], Renata Oliveira Neves[1‡], Paula Ruffoni Moreira[2‡], Juliana Rombaldi Bernardi[1,2,3]

1 The Graduate Program in Child and Adolescent Health, School of Medicine, Universidade Federal do Rio Grande do Sul (UFRGS), Porto Alegre, RS, Brazil, 2 Hospital de Clínicas de Porto Alegre (HCPA), Porto Alegre, Brazil, 3 The Graduate Program in Food, Nutrition and Health, School of Medicine, Universidade Federal do Rio Grande do Sul (UFRGS), Porto Alegre, RS, Brazil

☯ These authors contributed equally to this work.
‡ These authors also contributed equally to this work.
* christy.sbelin@gmail.com

## Abstract

### Background

Infant-guided methods, such as Baby-Led Introduction to SolidS (BLISS), encourage children to feed themselves from the same food consumed by their family since the beginning of the introduction of complementary foods, in opposition to the Parent-Led Weaning (PLW) method, which proposes foods mashed with a fork and given by parents. Adherence to child-guided methods is low due to a lack of confidence in the children's ability to feed themselves. This study aimed to assess adherence to three methods of food introduction: PLW, BLISS, or mixed (PLW and BLISS) at seven, nine, and 12 months of age.

### Methods

A randomized clinical trial was conducted with mother-infant pairs undergoing intervention at 5.5 months of age. Data were presented in absolute numbers and percentages and analyzed using the Chi-Square test.

### Results

The sample was constituted of 139 mother-infant pairs: 45 (32%) used PLW, 48 (35%) used BLISS, and 46 (33%) used the mixed method. Adherence to the method at seven, nine, and 12 months of age children was 34.1% (n = 45), 28.5% (n = 37), and 34.1% (n = 46), respectively. The mixed method presented significantly higher adherence results: 69.0% (n = 29) at seven months, 55.8% (n = 24) at nine months, and 78.6% (n = 33) at 12 months (p<0.001). Among the sample that unfollowed the proposed method, those who used PLW and BLISS migrated mostly to the mixed method at 12 months, 60.0% (n = 27) and 72.9% (n = 35) of them, respectively, because of the feeding mode and 97.8% (n = 44) and 100.0% (n = 48) because of food consistency.

**Data Availability Statement:** The datasets generated and/or analyzed during the current study are not publicly available because they are part of a larger study, with analyzes still being carried out.

For this reason, the authors of this article have not released the data for public access on any online platform. If requested, the corresponding author can send them directly by email. The non-author point of contact is Prof. Dr Erissandra Gomes. Telephone number: +55 51 99152-5376. E-mail: erifono@hotmail.com.

**Funding:** This study was supported by the Research and Events Incentive Fund–Hospital de Clínicas de Porto Alegre (grant number 2019-0230) awarded to LMN. This study was also financed by the Conselho Nacional de Desenvolvimento Científico e Tecnológico (CNPq) and Ministério da Educação, both awarded to CHSB.

**Competing interests:** The authors have declared that no competing interests exist.

## Conclusion

Complementary feeding in a mixed method presented higher adherence at seven, nine, and 12 months of age of children, which shows the feasibility of this approach to guide families in the introduction of complementary feeding.

## Introduction

Complementary feeding (CF) is the process of introducing solid and liquid foods, except for breast milk or infant formula, into an infant's diet [1]. The CF period is fundamental for all stages of life, especially the early years, which are decisive for the growth and development of children, the formation of eating habits, and health maintenance [2]. The Brazilian Ministry of Health recommends offering soft foods in large pieces for children to take to their mouths, and these foods may be initially mashed with a fork or chopped until it gradually reaches the consistency of the family's food at 12 months of age children [2].

In recent decades, responsive CF methods, such as Baby-Led Weaning (BLW) and Baby-Led Introduction to SolidS (BLISS), have been proposed [2,3]. Unlike the Parent-Led Weaning (PLW) method, which guides the initial consumption of foods mashed with a fork or chopped until it gradually reaches the consistency of the family's food, in child-guided methods, caregivers supervise children but do not take food to their mouths, allowing children to feed themselves on the same meal consumed by their family from the beginning of CF, as long as the food is in safe formats and consistencies [3]. BLISS emerged as a modified version of BLW, to prevent choking and adjust calories and iron intake [4].

Encouraging children's autonomy [5], increasing the consumption of vegetables and protein-rich foods [6], decreasing children's fussiness during meals [7,8], and increasing satiety responsiveness [9] are some of the many benefits of responsive CF methods [10]. However, families' adherence to the responsive CF methods is still low, as shown by a study carried out in Spain with 6,355 mother-child pairs [11]. Confidence in the children's ability to feed themselves, difficulty in measuring food intake, mess, and waste at mealtimes, and the possibility of choking are frequent concerns among mothers who use BLW or BLISS to feed their children [12,13]. Thus, they choose to use different CF methods simultaneously, also known as mixed methods, to feed their children, sometimes giving them porridge and purees with a spoon, and letting children feed themselves with food cut into sticks or strips [14].

The positive perceptions presented by studies regarding parents' adherence to responsive CF methods show that this method is easier and healthier [14]. As children are included in the family meal and eat the same food as their family, this method is easier, faster, less stressful, and more pleasurable [11,14]. Moreover, children can eat what they need and when they need it [14], which would prevent stress at mealtimes [11].

The fear of children choking [15], the difficulty in knowing which foods to offer and when to offer them, and the dietary iron intake [16] are some factors that prevent parents from adhering to child-guided methods. Thus, measuring adherence to methods of food introduction is essential to assess whether families randomly assigned to the methods can adhere to them and formulate public policies involving child-guided methods of food introduction [16].

Due to the popularity of responsive CF methods, their benefits, and the challenges for their adherence by families, this study aimed to assess adherence to three different CF methods at seven, nine, and 12 months of age of children: PLW, BLISS, and mixed, which is the combination of the BLISS method with the PLW method.

## Materials and methods

This is a randomized clinical trial with three intervention arms according to the CF method: PLW (control group), BLISS, and mixed, created especially for this study [17,18].

The sample was constituted of pairs of mothers and their children invited via social networking pages (internet groups of mothers) and a widely circulated newspaper in the city. In the invitation letter, a phone number and an email address were written so that parents could leave a message if they wanted to participate. Considering the inclusion criteria, those considered eligible received a standardized message explaining the study details, its risks, and benefits, and any additional questions were answered by the researchers by phone or email. A standardized text explained the stages of the study, the intervention at 5.5 months of age of children, the call at seven months, the home visit at nine months, and the face-to-face data collection at 12 months at the hospital.

Mothers living in the city of Porto Alegre, Southern Brazil, or its metropolitan region with healthy, full-term, non-twin children, birth weight $\geq$ 2500 g; age range < six months of life, and who had not started the complementary feeding process yet, were included. Initially, children with congenital malformations, neurological deficits, or some type of food restriction in the first year of life would be excluded from the study, but no such cases were observed during recruitment.

Participants were sequentially numbered and included in an allocated group previously computer-generated three-block randomization list (http://www.randomization.com) created by a researcher not involved in the intervention or data collection. Mothers only knew the group to which they were allocated on the day of the first intervention.

## Intervention

The intervention sessions were conducted in groups of up to eight pairs of mothers and/or caregivers and children, according to the CF randomized method (PLW, BLISS, or mixed).

The first intervention took place when the children were 5.5 months old in a private nutrition clinic equipped with an experimental kitchen and a classroom. They were performed by a team consisting of two nutritionists, who provided detailed guidance on the CF method of each group.

During a one-hour meeting, a nutritionist cooked examples of dietary foods and explained standardized information about the CF method to which participants were randomized. During the 'first 12 months of the life of children, the nutritionist's phone number and email address would be available to provide any extra support related to the CF method or for mothers to report adverse events. Flyers with additional information about CF and examples of healthy foods were also provided.

This study occurred in the transition period between information from the 2015 notebook 23 on breastfeeding and complementary feeding and the new information from the 2019 Dietary Guide for Brazilian Children Under 2 Years of Age, therefore, guidelines regarding food consistency for the group using PLW were those recommended by the notebook 23 [19]. The Brazilian Ministry of Health follows the guidelines of the World Health Organization on breastfeeding (BF), exclusive breastfeeding (EBF), and CF [20].

Regarding PLW, guidelines addressed the importance of EBF for up to six months of age of children and starting a slow and gradual introduction of complementary foods (cereals, tubers, meats, fruits, and vegetables) three times per day, without rigid schedules and respecting the children's appetite, when children are six months old, increasing this offer as the months go by and maintaining breast milk up to two years of age or more. Food consistency should be initially (from six to eight months of age) mashed with a fork until it gradually reaches the

consistency of the family's food at 12 months of age, with a variety of colors and food groups at every meal, without mixing and/or sieving the food. Food preparations should be separated so that children assimilate the flavors and characteristics of each one. Parents should not offer low-energy density preparations, such as soups and broths, besides sugar, coffee, canned goods, fried foods, soft drinks, juices, candies, snacks, and others in the first two years of life [2].

For BLISS, guidelines on BF were the same as for PLW, but on CF, parents should encourage their children to feed themselves-although always be assisted by an adult, and participate in family meals. Food from six months of age should be natural and cut into sticks or strips, instead of rounded shapes, for example. These formats allow children to eat with their own hands, making food easier to be pinched by children's 'fingers and preventing choking. Parents should avoid rushing children, respecting their time to explore flavors and textures while eating. The variety and quality of food in this method are the same as in PLW, as well as the notions of hygiene and the use of salt (8). Parents were also encouraged to offer three types of food at each meal: a food source of iron (for example, red meat), a food source of energy, and a food source of fiber, such as fruits or vegetables [21].

As for the mixed method, BF recommendations were the same as for the two methods mentioned before, but concerning CF, mothers should initially use BLISS. If children were not satiated or interested in foods offered according to BLISS, foods were offered according to PLW at the same meal.

Regardless of the group of which they were part, all mothers received guidance on how to avoid and manage possible choking during feeding. The caregivers were also trained and received printed material on how to perceive the child's satiety signals for all three methods.

## Data collection

Sociodemographic, family and child variables, maternal age (years), maternal schooling (years), family income (dollar), maternal ethnicity (white or non-white), cohabitation with a partner (yes or no), parity (primiparous or multipara), number of prenatal consultations (continuous), type of delivery (vaginal or cesarean section), child's gender (female or male) and EBF at six months (yes or no), were asked using an online form.

At 12 months, a questionnaire with four questions was applied regarding feasibility, acceptability, cost, and if the mother considered the method appropriate for her child.

EBF was defined as when the child received no liquid or solid other than human milk–not even water–except the oral rehydration solution, or drops/syrups of vitamins, minerals, or medications [20].

## Adherence to methods of food introduction

Follow-up data on family adherence to the CF method were collected until the end of the first year of life when children were seven, nine, and 12 months old.

The interview was conducted by a researcher blinded to the method using a standardized questionnaire designed for this study on food consistency and form of offering. It was questioned whether most of the time the food was liquidized, scraped, mashed, sieved, or cut into strips and of the same consistency as the family's food. Regarding the form of offer, those responsible answered whether the children took the food to their mouths alone or if they received complementary foods by spoon-feeding them up. Answers were provided as yes or no.

At the 7-month interview, by phone, at nine months of age, home visits, and 12 months of age, at the HCPA Clinical Research Center (CRC), Porto Alegre.

During the COVID-19 pandemic, at 9 months 50.7% ($n$ = 67) of the sample answered the questionnaires online, and at 12 months 80.3% ($n$ = 94) answered the online version of the same questionnaires.

Adherence to the method was separated by feeding mode (if children received food from a spoon or took it to their mouths with their hand) and food consistency (if the food was liquidized, scraped, mashed, sieved, or cut into strips and the same consistency of the family's food). Adherence to PLW was represented by parents who offered food with a spoon (feeding mode) and of a liquidized, scraped, mashed, or sieved consistency (food consistency) most of the time. Adherence to BLISS was represented by children who took food (feeding mode) of the same consistency as their family's or in strips to their mouths (food consistency), and adherence to the mixed method was represented by those who used alternating feeding modes and consistencies most of the time. Most of the time was defined as the child's usual diet or 75% of meals.

## Sample size

The sample size was estimated for the main outcome of the randomized clinical trial, child body mass index (BMI) for ages (17, 18), using WinPepi® software version 11.65, based on previously published studies on the subject. Considering a unit standard deviation of 1, the statistical power of 80%, and a significance level of 5% to detect a difference in BMI of 0.8 kg/m$^2$, the sample calculation for a half standard deviation difference will consist of 48 mother-infant pairs for each of the three intervention groups, totaling 144 pairs of mothers and infants. For this secondary outcome, the total sample size was used.

## Statistical analysis

The database was created using double data entry. Statistical analyses were performed using the software Statistical Package for the Social Sciences® (SPSS, Inc., Chicago, IL, USA) version 22.0 for Windows. A descriptive analysis of continuous and categorical variables was performed. Qualitative characteristics were expressed as absolute numbers and percentages and Pearson's chi-square test was used to assess differences in proportions. Parametric data were expressed as mean ± standard deviation (SD) and analyzed by Fisher's exact test and ANOVA with Bonferroni's post hoc test. Nonparametric data were expressed as median and interquartile ranges and analyzed using the Kruskal-Wallis test with Dunn's post hoc test. To assess differences between variables with normal distribution between groups of children, ANOVA and Bonferroni post hoc tests were used. To analyze factors associated with adherence results, Poisson regression was used. The chi-square test was used to assess differences in proportions between different groups. A 5% significance level ($p < 0.05$) and a 95% confidence interval were considered.

## Ethical aspects

All responsible parties first signed the informed consent form online and, after that, they signed it in person in two copies, one for the research team and one for the child's guardian.

This study was approved by the Research Ethics Committee of the Hospital de Clínicas de Porto Alegre under number 2019–0230, CAAE: 1537018500005327, and is registered in the Brazilian Clinical Trials Registry (RBR-229SCM).

## Results

The flowchart of the participant selection process for the randomized clinical trial from the recruitment of mother-child pairs to the assessment performed when children were 12 months old was presented in Fig 1.

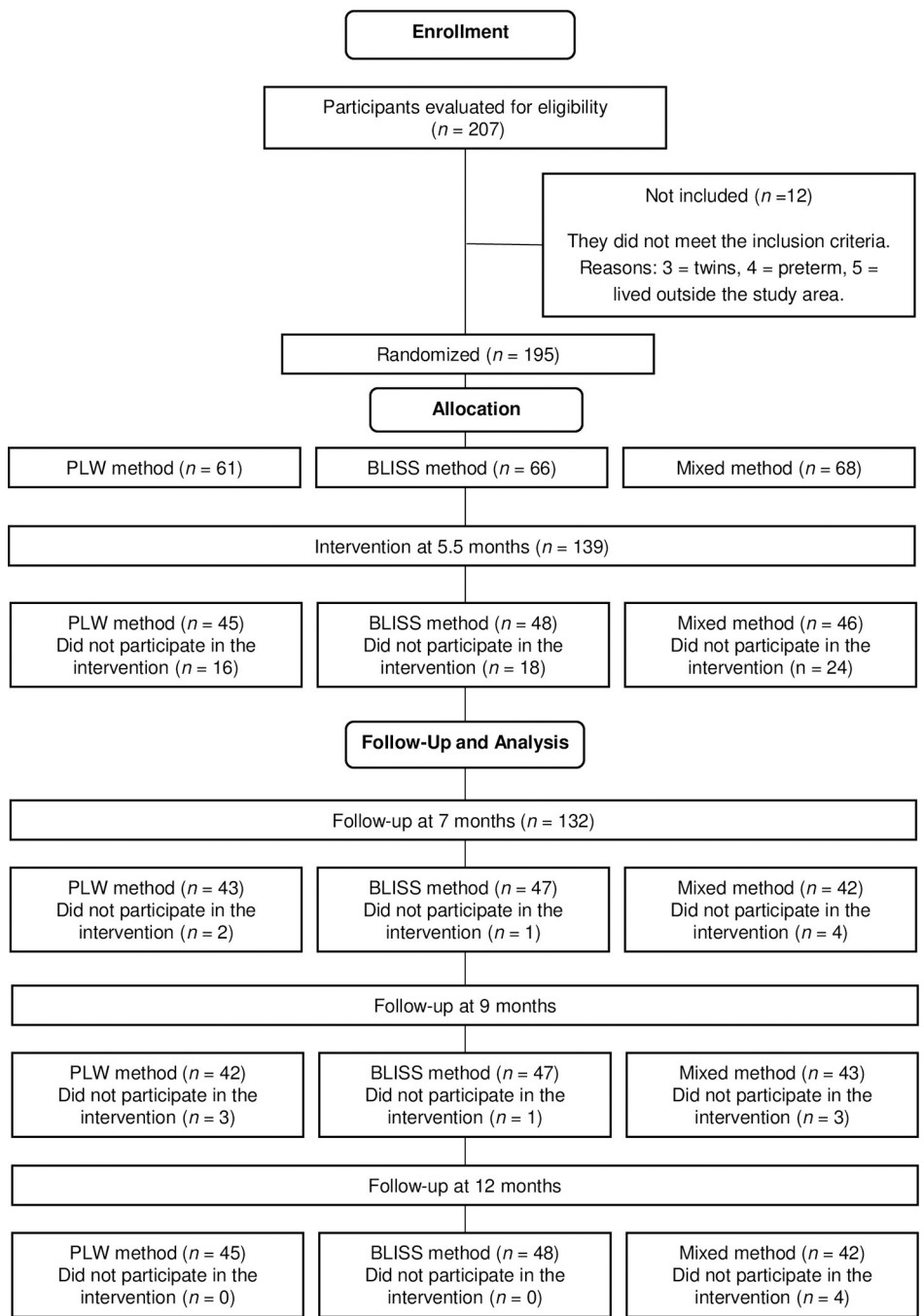

**Fig 1. Study design flowchart.** PLW: Parent-Led Weaning, BLISS: Baby-Led Introduction to SolidS.

In total, 207 mother-child pairs contacted the research team, of which 12 (5.7%) did not meet the inclusion criteria, thus only 195 mother-infant pairs were eligible for the study. Among these 195 pairs, 56 did not participate in the intervention, totaling a sample of 139 mother-child pairs.

After randomization regarding the feeding methods, 45 pairs (32%) were submitted to PLW, 48 (35%) to BLISS, and 46 (33%) to the mixed method. Most mothers declared themselves white (86.2%; $n$ = 119), with a mean age of 32.3 ($\pm$5.27) years old, a mean of 17.68

(±5.28) years of schooling, and a median monthly household income of U$1,586.00 [1,015.00–2,538.00], with no significant differences between groups for both household income (p = 0.277) and maternal schooling (p = 0.457). The three groups also presented no statistically significant differences regarding maternal age, maternal schooling, ethnicity, total household income, cohabitation with a partner, parity, number of prenatal visits, type of delivery, sex of children, and BF up to six months of age (p>0.05). Table 1 shows the sample characteristics of methods of food introduction.

Feeding questionnaires were completed by 132 mother-child pairs at seven and nine months of age of children and 135 mother-infant pairs at 12 months. Of the answered questionnaires, 50% (n = 67) were applied online at nine months and 92.8% (n = 104) at 12 months.

Table 2 shows the values of adherence to methods separated by feeding mode and consistency. Total adherence to feeding mode and food consistency of the methods proposed at seven, nine, and 12 months was 34.1% (n = 45), 28.5% (n = 37), and 34.1% (n = 46), respectively. The mixed method presented significantly higher values than the others, with adherence of 69.0% (n = 29) at seven months, 55.8% (n = 24) at nine months, and 78.6% (n = 33) at 12 months, followed by PLW (p<0.001).

Analyzing feeding mode separated from food consistency, at seven and 12 months, families that adhered to the proposed method used more the consistency of the mixed method (prevalence of 19.0%; n = 8). Children initially submitted to PLW showed lower adherence to the method at seven, nine, and 12 months of age than those randomized to the mixed method. At 12 months, when asked about the feasibility, acceptability, and cost of the CF method, as well as if it was appropriate for their child, 94.1% (n = 127) of mothers answered that the CF method to which they were randomized was feasible, 91.1% (n = 123) that the level of mess

**Table 1. Characterization of the sample of mothers and children from Porto Alegre and its metropolitan region, Rio Grande do Sul, Brazil, 2021.**

| Family variables | Total mean (± SD), median (IR) | PLW (n = 45) Mean (± SD) Median (IR) | BLISS (n = 48) Mean (± SD) Median (IR) | Mixed (n = 46) Mean (± SD) Median (IR) | p-value |
|---|---|---|---|---|---|
| Maternal age (years) | 32.3 (5.27) | 31.6 (5.59) | 33.65 (5.20) | 31.68 (4.6) | 0.113* |
| Maternal schooling (years) | 17.68 (5.28) | 16.88 (5.04) | 18.28 (6.24) | 17.80 (4.35) | 0.457[1] |
| Family income (USD) | 1.584,24 [1.013,9–2.534,7] | 1.267,3 [975,8–2.534,7] | 2.027,8 [1.013,9–3.421,9] | 1.520,8 [950,5–2.534,7] | 0.277[2] |
| Maternal ethnicity white | 119 (86.2) | 37 (84.1) | 41 (85.4) | 41 (89.1) | 0.765[3] |
| Cohabitation with a partner | 117 (84.2) | 35 (77.8) | 43 (89.6) | 39 (84.8) | 0.295[3] |
| Primiparity | 113 (81.3) | 35 (77.8) | 37 (77.1) | 41 (89.1) | 0.223[3] |
| Prenatal visits (number) | 11.26 (2.54) | 11.23 (2.42) | 11.31 (2.75) | 11.24 (2.48) | 0.985[1] |
| Cesarean section | 85 (61.2) | 33 (73.3) | 29 (60.4) | 23 (50.0) | 0.760[3] |
| Infant variables | Total mean (± SD), median (IR) | PLW (n = 45) Mean (± SD) Median (IR) | BLISS (n = 48) Mean (± SD) Median (IR) | Mixed (n = 46) Mean (± SD) Median (IR) | p-value |
| Child's gender female | 72 (51.8) | 25 (55.6) | 26 (54.2) | 21 (45.7) | 0.589[3] |
| Exclusive breastfeeding up to six months of age | 113 (85.3) | 38 (90.5) | 38 (80.9) | 37 (86.0) | 0.413[3] |

PLW: Parent-Led Weaning; BLISS: Baby-Led Introduction to SolidS; SD: Standard deviation; IR: Interquartile range; USD: United States Dollar.

*Fisher's exact test

[1] ANOVA with Bonferroni post hoc test

[2] Kruskal-Wallis test with Dunn's post hoc test

[3] Pearson's chi-square test.

**Table 2. Adherence to methods separated by feeding mode and consistency.**

| Variables | | PLW n (%) | BLISS n (%) | Mixed n (%) | Total n (%) | p-value |
|---|---|---|---|---|---|---|
| The randomized method at 7 months of age | Does not follow the method | 26 (60.5)[a] | 38 (80.8)[a] | 4 (9.5)[b] | 68 (51.5) | < 0.001[1] |
| | Follows mode only | 5 (11.6)[a] | 2 (4.3)[a] | 1 (2.4)[a] | 8 (6.1) | < 0.001[1] |
| | Follows consistency only | 1 (2.3)[a] | 2 (4.3)[ab] | 8 (19.0)[b] | 11 (8.3) | < 0.001[1] |
| | Follow both | 11 (25.6)[a] | 5 (10.6)[a] | 29 (69.0)[b] | 45 (34.1) | < 0.001[1] |
| The randomized method at 9 months of age | Does not follow the method | 22 (55.0)[a] | 32 (68.1)[a] | 5 (11.6)[b] | 59 (45.4) | < 0.001[2] |
| | Follows mode only | 8 (20.0)[a] | 1 (2.1)[b] | 8 (18.6)[a] | 17 (13.1) | < 0.001[2] |
| | Follows consistency only | 0[a] | 11 (23.4)[b] | 6 (14.0)[b] | 17 (13.1) | < 0.001[2] |
| | Follow both | 10 (25.0)[a] | 3 (6.4)[b] | 24 (55.8)[c] | 37 (28.5) | < 0.001[2] |
| The randomized method at 12 months of age | Does not follow the method | 32 (71.1)[a] | 40 (83.3)[a] | 0[b] | 72 (53.3) | < 0.001[1] |
| | Follows mode only | 0[a] | 8 (16.7)[b] | 1 (2.4)[ab] | 9 (6.7) | < 0.001[1] |
| | Follows consistency only | 0[a] | 0[a] | 8 (19.0)[b] | 8 (5.9) | < 0.001[1] |
| | Follow both | 13 (28.9)[a] | 0[b] | 33 (78.6)[c] | 46 (34.1) | < 0.001[1] |

PLW: Parent-Led Weaning; BLISS: Baby-Led Introduction to SolidS.

[1]Fisher's exact test

[2]Pearson's chi-square test.

* Different letters point to statistically significant differences between groups.

was acceptable, 92.6% ($n = 125$) that the cost was affordable, and 94.1% ($n = 127$) that they considered the method appropriate for their child. However, mothers using PLW ($n = 45$; 100.0%) and the mixed method ($n = 41$; 97.6%) presented a statistically significant difference in comparison with mothers using BLISS regarding the feasibility of the method ($p = 0.013$) and if it was appropriate for their child ($p = 0.011$).

Figs 2 and 3 show the behavior of children regarding feeding method, referred, and food consistency at seven, nine, and twelve months of age. The curves demonstrate that the mixed method was the most followed in consistency and feeding method, followed by the PLW method, and the method that was least followed at all times was the BLISS method.

Regarding the feeding mode (Fig 2), at seven months, 92 (66.2%) children were following the mixed method, 28 (20.1%) were following the PLW method, and 12 (8.6%) were following

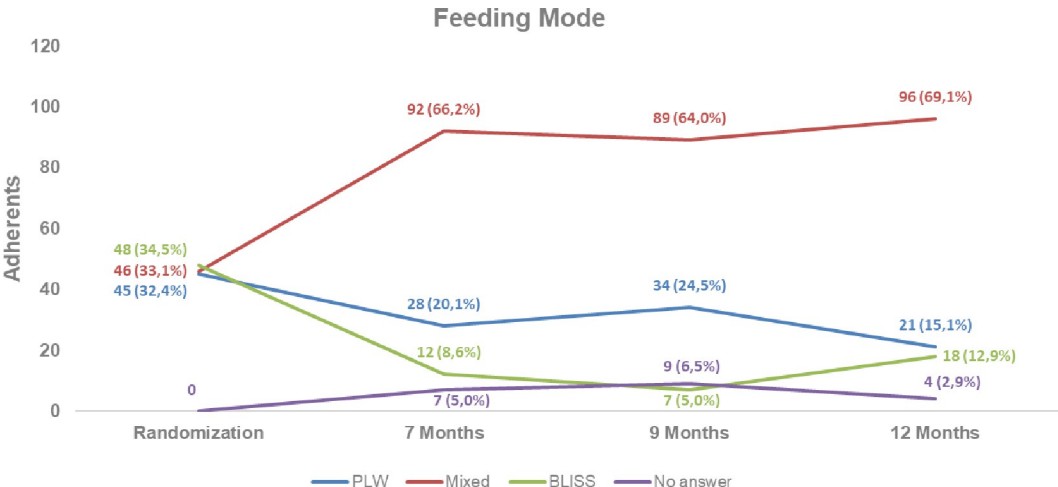

**Fig 2. Adherence to the feeding mode at seven, nine, and 12 months of life of children per randomized group.** PLW: Parent-Led Weaning; BLISS: Baby-Led Introduction to SolidS.

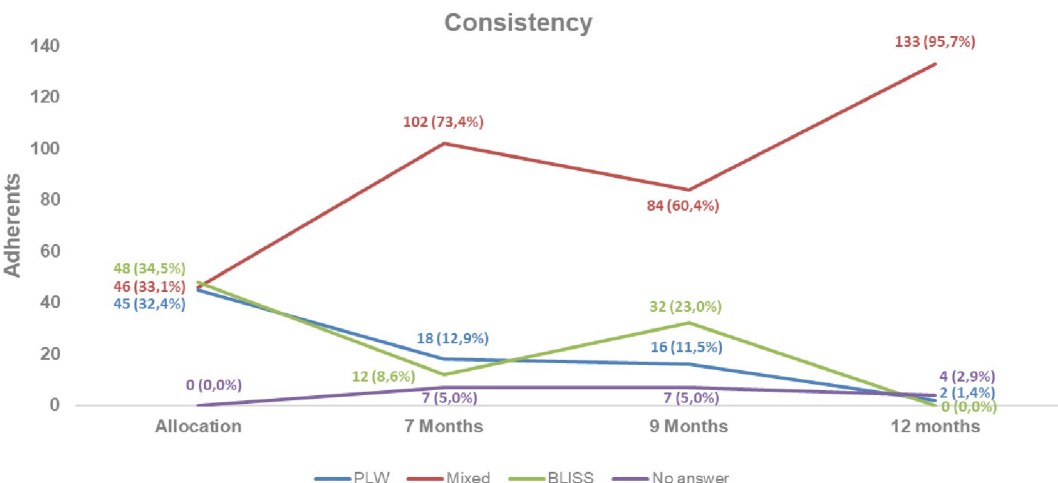

**Fig 3. Adherence to the consistency at seven, nine, and 12 months of life per randomized group.** PLW: Parent-Led Weaning; BLISS: Baby-Led Introduction to SolidS.

the BLISS method. At nine months, 89 (64.4%) children were following the mixed method, 34 (24.5%) were following the PLW method, and 9 (6.5%) were following the BLISS method. At 12 months, 96 (69.1%) children were following the mixed method, 21 (15.1%) were following the PLW method, and 18 (12.9%) were following the BLISS method. The mixed method was the most followed by the children, and the BLISS method was followed by the least children at all evaluated ages (seven, nine, and 12 months).

Regarding the consistency of feeding (Fig 3), at seven months 102 (73.4%) children were following the mixed method, 18 (12.9%) were following the PLW method, and 12 (8.6%) were following the BLISS method. At nine months, 84 (60.4%) children were following the mixed method, 16 (11.5%) were following the PLW method, and 32 (23.0%) were following the BLISS method. At 12 months, 133 (95.7%) children were following the mixed method, 2 (1.4%) were following the PLW method, and none were following the BLISS method.

Tables 3 and 4 show the behavior of adherence and migration of pairs according to the randomized group regarding feeding mode and food consistency. The mixed method presented

**Table 3. Feeding mode migration among those who did not follow the proposed method at seven, nine, and 12 months of age of children per randomized group.**

| Randomized method | Migrated Group | 7 months _n_ (%) | 9 months _n_ (%) | 12 months _n_ (%) |
|---|---|---|---|---|
| **PLW Group** | No answer | 2 (4.4) | 5 (11.1) | 0 (0.0) |
| | PLW | 16 (35.6) | 18 (40.0) | 13 (28.9) |
| | Mixed | 25 (55.6) | 20 (44.4) | 27 (60.0) |
| | BLISS | 2 (4.4) | 2 (4.4) | 5 (11.1) |
| **Mixed Group** | No answer | 4 (8.7) | 3 (6.5) | 4 (8.7) |
| | Mixed | 30 (65.2) | 32 (69.6) | 34 (73.9) |
| | PLW | 9 (19.6) | 10 (21.7) | 3 (6.5) |
| | BLISS | 3 (6.5) | 1 (2.2) | 5 (10.9) |
| **BLISS Group** | No answer | 1 (2.1) | 1 (2.1) | 0 (0.0) |
| | BLISS | 7 (14.6) | 4 (8.3) | 8 (16.7) |
| | PLW | 3 (6.3) | 6 (12.5) | 5 (10.4) |
| | Mixed | 37 (77.1) | 37 (77.1) | 35 (72.9) |

PLW: Parent-Led Weaning; BLISS: Baby-Led Introduction to SolidS.

**Table 4. Consistency migration among those who did not follow the proposed method at seven, nine, and 12 months of age of children per randomized group.**

| Randomized method | Migrated Group | 7 months<br>*n* (%) | 9 months<br>*n* (%) | 12 months<br>*n* (%) |
|---|---|---|---|---|
| **PLW Group** | No answer | 2 (4.4) | 3 (6.7) | 0 (0.0) |
| | PLW | 12 (26.7) | 11 (24.4) | 1 (2.2) |
| | Mixed | 28 (62.2) | 24 (53.3) | 44 (97.8) |
| | BLISS | 3 (6.7) | 7 (15.6) | 0 (0.0) |
| **Mixed Group** | Mixed | 4 (8.7) | 3 (6.5) | 4 (8.7) |
| | Follow | 37 (80.4) | 30 (65.2) | 41 (89.1) |
| | PLW | 3 (6.5) | 2 (4.3) | 1 (2.2) |
| | BLISS | 2 (4.3) | 11 (23.9) | 0 (0.0) |
| **BLISS Group** | BLISS | 1 (2.1) | 1 (2.1) | 0 (0.0) |
| | Follow | 7 (14.6) | 14 (29.2) | 0 (0.0) |
| | PLW | 3 (6.3) | 3 (6.3) | 0 (0.0) |
| | Mixed | 37 (77.1) | 30 (62.5) | 48 (100.0) |

PLW: Parent-Led Weaning; BLISS: Baby-Led Introduction to SolidS.

the lowest number of migrations to other methods and was the main option for pairs who migrated from PLW and BLISS. Concerning the feeding mode, at 12 months of age, 73.9% of children initially randomized to this method continued to follow it. For PLW and BLISS, respectively, 28.9% and 16.7% of their initial sample number remained and 60% and 72.9% migrated to the mixed method. Regarding food consistency, 89.1% of pairs using the mixed method continued to follow it but only 2.2% of the pairs continued using PLW and 97.8% of its randomized group migrated to the mixed method. BLISS presented no adherents, as 100% of its initially randomized group migrated to the mixed method.

Regarding factors related to adherence to CF methods, adherence to methods at seven, nine, and 12 months of age of children presented no statistically significant association between family and maternal variables, such as maternal skin color, age, schooling, type of delivery, family income, cohabitation with a partner, and daily working hours, and infant variables, such as sex of children, BF at hospital discharge, and exclusive breastfeeding up to their six months of age (Table 5).

## Discussion

The results of this study demonstrate that the mixed method of complementary feeding, which includes offering foods cut into sticks or strips and mashed or pureed foods, had higher adherence in our sample of children at seven, nine, and 12 months of age. Additionally, the mixed method was followed by most of the children who did not adhere to the CF methods they were randomized into. In this sample, the BLISS method had the lowest adherence every month. Furthermore, the authors did not find any association between adherence to the introduction of food methods and maternal, familial, and infant variables analyzed.

Concerning the three-group randomization, the authors observed no significant association between household income, maternal age, schooling level, parity, and BF up to six months of age of children. The authors [7] showed that the use of PLW and BLISS presented no differences regarding maternal age (mean 31.3 years), skin color (81.6% Caucasian), and parity (41.3% were primiparous).

The authors [15] found that, in clinical practice, 8% of the sample adhered to BLW, in which children feed themselves always or most of the time at six and seven months of age.

**Table 5. Association between adherence to methods of food introduction and maternal and infant variables.**

| Variables | Prevalence ratio (95% CI)– 7 months | Prevalence ratio (95% CI)– 9 months | Prevalence ratio (95% CI)– 12 months | Prevalence ratio (95% CI) | p-value |
|---|---|---|---|---|---|
| *Group of which they were part* | | | | | |
| PLW | 0.43 (0.29–0.64) | 0.50 (0.35–0.72) | 0.28 (0.18–0.45) | - | < 0.001 |
| BLISS | 0.21 (0.11–0.38) | 0.36 (0.23–0.55) | 0.16 (0.08–0.31) | - | < 0.001 |
| Mixed | 1 (reference) | 1 (reference) | 1 (reference) | - | - |
| *Maternal variables* | | | | | |
| Maternal skin color | - | - | - | 0.744 (0.30–1.80) | 0.512 |
| Maternal age 25–35 years old | - | - | - | 2.4 (0.65–8.88) | 0.187 |
| Cesarean section | - | - | - | 0.72 (0.44–1.16) | 0.184 |
| Maternal schooling level (over 11 years of schooling) | - | - | - | 1.16 (0.36–3.72) | 0.795 |
| Total household income (up to six minimum wages) * | - | - | - | 1.39 (0.65–2.94) | 0.386 |
| Cohabitation with a partner | - | - | - | 0.56 (0.22–1.40) | 0.219 |
| Primiparity | - | - | - | 1.22 (0.61–2.41) | 0.561 |
| Works outside the home (up to six hours/day) | - | - | - | 1.20 (0.76–1.89) | 0.431 |
| *Infant variables* | | | | | |
| Exclusive breastfeeding at hospital discharge | - | - | - | 0.79 (0.43–1.45) | 0.458 |
| Exclusive breastfeeding up to six months of age | - | - | - | 0.91 (0.44–1.85) | 0.804 |

PLW: Parent-Led Weaning; BLISS: Baby-Led Introduction to SolidS; CI: Confidence interval. p-values were calculated by Poisson regression.

Moreover, 21% of families reported following BLW, however, they followed a method that combined children's 'self-feeding and food offered with a spoon [12]. In our study, at seven months of age of the children, 77.1% of the pairs randomized to the BLISS method were following the mixed method. However, another study showed that adherence to BLISS was higher, presenting significantly more children eating most or all of their food in the week before turning seven and 12 months of age [22]. This difference is probably due to the popularity of child-guided methods, which present a higher prevalence in the United Kingdom, the United States, New Zealand, and Canada [21]. To date, no studies assess the prevalence of BLW or BLISS among the Brazilian population.

Adherence to BLISS was the lowest within our sample, in opposition to [16], whose study showed that 40%, 67%, and 96% of children from her sample fed themselves at seven, 12, and 24 months, respectively. The authors assessed adherence to BLISS by calculating the percentage of daily food intake by weight (g) consumed by children who fed themselves, fed themselves, and were fed by an adult, or were only fed by an adult [16]. Unlike other studies [9,11], our study chose to assess adherence to methods separated by feeding mode and food consistency due to the alternation in feeding mode (caregivers offering food with a spoon or allowing children to feed themselves) and food consistency (foods mashed with a fork and foods cut into sticks or strips) [23] used questionnaires to assess families' adherence to BLISS compared to PLW (control), and parents submitted to both methods reported high levels of satisfaction and convenience, which corroborates our data since mothers agreed that the BLISS method was viable, accessible, and suitable for their children, presenting an acceptable level of mess. According to study [4], the group randomized to BLISS reported less mess and a more expensive cost, but when analyzing food costs, the groups did not show significant differences. Thus, parents who used both BLISS and PLW considered the child-guided method acceptable [4].

Moreover, in our study, mother-child pairs either adhered to methods by feeding mode or food consistency, so the authors chose to separate them. Thus, it was the first study to separate feeding mode and food consistency in all months. The Brazilian Ministry of Health advises

that, at 12 months of age, children must be encouraged to take food with their own hands to stimulate the pinch movement and teach them to cut food with their front teeth [2,9]. When analyzing adherence to BLISS and PLW, results were separated into three groups: one in which children took all or most of the food to their mouths, another in which children were fed by their parents, and another in which they were equally fed by their parents and took food to their mouths [9]. In our sample, at 12 months of age, children either took food to their mouths (mashed with a fork or cut into sticks or strips) or were fed with a spoon by their parents.

Mothers using BLISS who did not follow the method migrated mainly to the mixed method, compared to a study by [24], in which mothers mixed foods cut into sticks or strips with food of other textures given with spoons to help children when they were unable to eat and avoid a mess [24].

Even though few studies assess the relationship between adherence to CF methods and maternal, family, or infant variables, the authors show that what would represent adherence to child-guided methods, identifying a cultural variable related to adherence to a combination with PLW, lacks definition. The literature does not clearly show how common the combination of PLW with child-guided methods is, as most studies ask participants to identify the specific method they are following [25], and even when BLW is objectively defined, some foods might be offered with spoons and as purees, [26] classified BLW user participants who used spoons and offered food in the form of porridge and purees less than 10% of the time. According to [10] in a cross-sectional study designed with 261 mothers who reported following BLW in Chile, 57,5% reported their child ate the same food as the family, and 75,6% reported only occasionally offering food with a spoon.

The authors found no association between adherence to methods of food introduction and family and maternal variables, such as maternal skin color and age, type of delivery, monthly household income, cohabitation with a partner, parity, daily working hours, and infant variables, such as BF at hospital discharge and BF up to six months of age of children. Our data is following [27], who also found no differences between BLISS and BLW concerning factors related to adherence to child-guided methods (proportion of self-feeding, food consumed by the family, and family meals shared with children) at six, seven, or eight months of age of children [27].

Other variables not analyzed in this study may influence adherence to CF methods. For example, parental behavior, as [28] showed in their study, which analyzed maternal eating behavior related to CF methods, from adherence to strict BLW (children feed themselves 90% of the time or more) to adherence to strict PLW (children feed themselves less than 10% of the time), to allow a more inclusive categorization than studies performed only with strict BLW. Their results showed that parents who followed strictly BLW were less controlling of food and less likely to use encouragement as a technique to increase their children's food intake. Moreover, they tended to offer more vegetables and have more meals with their children, as well as offer the same food they eat more often than parents who followed other methods [28].

Child-guided methods are associated with a low-control maternal feeding style, in which mothers control the children's food intake less than mothers who use PLW; thus, parenting style could be an important factor to influence adherence to CF methods [11]. The mothers' willingness to follow the recommendations of CF provided by healthcare professionals may limit adherence to the methods, as mothers report doing what was best for their children and not following a healthcare professional's guidance [29]. Additionally, mothers who fed their children using the BLW method reported lower anxiety, lower scores of obsessive-compulsive disorder, lesser food restrictions, and greater awareness than mothers who opted for the PLW/traditional method. However, these results are from observational studies and require further

investigation [30]. It is important to highlight that the present study focused mainly on the parents' decision to practice or change the method of complementary feeding.

This research has some limitations. Just under 30% of the mothers who met the inclusion criteria did not attend the intervention; these mothers had lower schooling and family income than those included. In Brazil, maternity leave lasts four months in most companies and our intervention occurred at 5.5 months of the children's lives, therefore, there may be mothers unable to participate in this study. Family income and schooling level in our sample were higher than in the general population, which compromises the generalizability of results. The sample was spontaneously recruited mainly from the target social networks, which may result in mothers previously interested in complementary feeding. The sample size may have been insufficient to obtain significant results in the tests performed to assess the association. Another limitation of the study concerns how adherence was asked of those responsible for categorical questions with yes or no answers that took into account how the child's diet was most of the time (75% of meals). There is no definition of what is the best way to measure adherence to CF methods, this can generate heterogeneity in the way researchers evaluate in their studies. The possibility to contact the nutritionist by phone or email wasn't evaluated for the adherence to the method in this study.

However, this study provides a significant and unprecedented contribution to the field of complementary feeding. As far as we know, it is the first study to evaluate adherence to the major feeding introduction methods described in the scientific literature at three different time points. The study design included personalized nutritional counseling and practical intervention, with the aid of an experimental kitchen equipped for food preparation. However, this study is not exhaustive on the topic of adherence to complementary feeding methods. The authors recommend further studies focusing on standardizing the way adherence is evaluated.

## Conclusion

The mixed method of CF showed greater adherence at seven, nine, and 12 months of age in children compared to the PLW and BLISS methods. In addition, among those children who did not adhere to the method they were randomized to, the majority followed the mixed method. Therefore, the mixed method appears to be a feasible approach to guide families in the introduction of CF. Family and maternal variables, such as family income, schooling, daily working hours, type of delivery, and infant variables, such as breastfeeding at hospital discharge and EBF until six months of age, were not associated with adherence to the mixed method. Further studies are needed to examine adherence to different methods of CF, with special attention to the strategies used to evaluate adherence. The authors also reinforce that CF is associated with cultural aspects of food, and the generalization of data should consider such aspects.

## Acknowledgments

The authors are grateful for the efforts of all researchers in the study and the willingness of all participants.

## Author Contributions

**Conceptualization:** Christy Hannah Sanini Belin, Leandro Meirelles Nunes, Juliana Rombaldi Bernardi.

**Data curation:** Christy Hannah Sanini Belin, Paula Ruffoni Moreira.

**Formal analysis:** Christy Hannah Sanini Belin, Renata Oliveira Neves, Paula Ruffoni Moreira.

**Funding acquisition:** Juliana Rombaldi Bernardi.

**Methodology:** Christy Hannah Sanini Belin, Leandro Meirelles Nunes, Renata Oliveira Neves, Juliana Rombaldi Bernardi.

**Project administration:** Juliana Rombaldi Bernardi.

**Supervision:** Leandro Meirelles Nunes, Juliana Rombaldi Bernardi.

**Visualization:** Cátia Regina Ficagna, Paula Ruffoni Moreira.

**Writing – original draft:** Christy Hannah Sanini Belin, Renata Oliveira Neves, Paula Ruffoni Moreira, Juliana Rombaldi Bernardi.

**Writing – review & editing:** Christy Hannah Sanini Belin, Leandro Meirelles Nunes, Cátia Regina Ficagna, Renata Oliveira Neves, Paula Ruffoni Moreira, Juliana Rombaldi Bernardi.

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
