## [Decision Letter · Decision Letter 0]

5 Apr 2023

PONE-D-22-20911ADHERENCE TO DIFFERENT COMPLEMENTARY FEEDING METHODS IN THE FIRST YEAR OF LIFE: A RANDOMIZED CLINICAL TRIALPLOS ONE

Dear Dr. Sanini Belin,

Thank you for submitting your manuscript to PLOS ONE. After careful consideration, we feel that it has merit but does not fully meet PLOS ONE’s publication criteria as it currently stands. Therefore, we invite you to submit a revised version of the manuscript that addresses the points raised during the review process.

 Please address the feedback as provided by both the reviewers below.

We look forward to receiving your revised manuscript.

Kind regards,

Erica Jane Cook, Ph.D

Academic Editor

PLOS ONE

Journal Requirements:

4. Please update your submission to use the PLOS LaTeX template. The template and more information on our requirements for LaTeX submissions can be found at http://journals.plos.org/plosone/s/latex.

Reviewers' comments:

Reviewer's Responses to Questions

**Comments to the Author**

1. Is the manuscript technically sound, and do the data support the conclusions?

Reviewer #1: Partly

Reviewer #2: Partly

2. Has the statistical analysis been performed appropriately and rigorously? 

Reviewer #1: Yes

Reviewer #2: Yes

3. Have the authors made all data underlying the findings in their manuscript fully available?

Reviewer #1: Yes

Reviewer #2: Yes

4. Is the manuscript presented in an intelligible fashion and written in standard English?

Reviewer #1: Yes

Reviewer #2: Yes

5. Review Comments to the Author

Reviewer #1: This study aimed to assess adherence to three methods of food introduction: PLW, BLISS, or mixed (PLW and BLISS) at seven, nine, and 12 months of age.The topic is of interest as the adherence to different methods of weaning have not been investigated yet. My comments, as following:

Introduction:

- In addition to the Brazilian recommendations, also the WHO recommendations for complementary feeding should be cited.

Materials and methods

“Each group was constituted of about eight pairs”: in the Results section, much more pairs are described. Please check this data

Intervention (5.5 months of age):

In the paragraph “Regarding PLW, …. without mixing and/or sieving the food. Food preparations should be separated so that children assimilate the flavors and characteristics of each one.”

I was quite puzzled to read this stament; one of the main characteristic of the traditional purèed infant foods is to be composed of several food items mixed together (e.g. cereal creme + homogenized meat + vegetables). Why did authors choose to propose foods separately? Could they explain this choice?

Data collection

“The interview was carried out by a researcher blinded to the method with a food frequency questionnaire and a food record prepared for this study.”: How was the food record performed? Please add some details

Results:

Results show that a substantial percentage of the children migrated from PLW/BLISS to mixed method (up to almost 100% both for PLW and BLISS with regard to consistency).

However, infants were followed from the seventh until the 12th month of life. This is a crucial stage in the infant's neuromotor development, and it is therefore reasonable to assume that during these months the feeding pattern may change. Could the authors add a comment on this in the discussion?

Discussion

- In the discussion authors say "mother-child pairs either adhered to methods by feeding mode or food consistency, so we chose to separate them."

However, one of the key differences between a traditional PLW and BLISS is indeed the consistency of the foods offered.

If authors aim to mantain a separate analysis for the feeding mode and consistency, I believe the reason of this distinction should be better clarified.

- In the paragraph “Child-guided method …. The mothers’ will to follow CF recommendations provided by health professionals was another limiting factor ...”:

in addition, more anxious mothers may more likely choose a traditional weaning approach such as PLW, where there is more existing data and support from healthcare professionals. This could be another important influencing factor, please add a comment in regard. In regard, authors may consider the systematic review by D'Auria E, et al. Baby-led weaning: what a systematic review of the literature adds on. Ital J Pediatr. 2018

Minor:

“an ANOVA test with Bonferroni post hoc test was used and.” : please complete the sentence

Introductiion: [14,11,14]: ref. 14 does not need to be repeated

and some minor typos need to be checked

Reviewer #2: Summary of the research

This study aimed to assess adherence to three methods of food introduction; parent led, baby led and mixed, with follow up at 7, 9 and 12 months. The intervention consisted of a one-hour group meeting with nutritionist who cooked examples of dietary foods and explained standardized information about the feeding method they had been randomised to. The study is described as a ‘randomised clinical trial’. Randomisation does indeed occur, specifically in to 3 treatment groups; parent led, baby led and mixed. However there is no control group (e,g., standard care, treatment as usual, no intervention).

Results suggest that mixed method of feeding results in higher adherence and most parents from the other two groups also migrated to this method. The authors conclude this shows the feasibility of a mixed approach to guide families in the introduction of complementary feeding. The discussion considers some but not all of the limitations of the study.

The research explores an important area, supplements previous work in the field and is novel in its exploration within a Brazilian sample. However, issues with the design, measurement and interpretation limit its value. There are some issues with details missing and clarity in expression.

My overall recommendation would be major revisions.

Examples and evidence

1. The study is described as a ‘randomised clinical trial’ and refers to parent-led weaning as a control. There is no control group (e.g., treatment as usual, standard care, no intervention); there are 3 treatment groups (parent led, child led and mixed) and no control. It could be argued that there is no need for a control group within this trial as the study explores ‘adherence’ and not outcomes, so there is no need for a baseline for determining the effectiveness of the study treatment. However, authors should think carefully about the design and methodology of their trial, describe it accurately and explain the rationale behind it. They should not make claims that is a RCT with control group, if it is not.

2. There are significant issues within the way adherence is measured, analysed and reported. For example on Page 6 the authors state that researchers ‘made a phone call and asked some questions, regarding food consistency and if children took food to mouth or received from mouth and another researcher analysed them’. What questions were they asked? How were responses interpreted, scored and analysed? At this stage it is unclear if responses were qualitative or quantitative, and if numerical responses were categorical (in which case what were they categories?) or on a continuum (again from what and to what?). In the discussion it becomes apparent that for feeding mode this was categorical with 3 groups (1. children took all or most of the food to their mouths; 2. children were fed by their parents; 3. they were equally fed by their parents and took food to their mouths). This needs to be much clearer within the methodology and also stated for feeding consistency. On page 8, Adherence to BLISS for consistency is described as ‘children who took food of the same consistency as their family’s or in strips to their mouths’ but it is not clear if this is all of the time, ‘all or most’ or a certain %. It is unclear if consistency adherence was also into 3 categories and if so, what those categories were.

This also brings to question the validity and quality of the measures used for adherence. A strength to this study is that it explores mode and consistency separately, however measuring adherence categorically, with poorly defined and reported categories, is a strong methodological flaw that really limits interpretation of the findings and the conclusions that can be drawn. What is meant by ‘all or most’? Previous studies (as noted in the discussion) define or measure this more tangibly/objectively. For example, Erickson et al., (2018) who assessed adherence to BLISS by calculating the percentage of daily food intake by weight (g) consumed by children who fed themselves/were fed by an adult. Komninou et al., (2019) classified adherence to strict BLW as children feeding themselves 90% of the time or more.

It is important that adherence to BLISS in this study is clearly defined so that results so be interpreted in a more meaningful way. Is adherence classed as only those categorised into the ‘all or most’ group, and none adherence is everyone in the 50/50 and fed by parents’ groups? Were participants only offered these 3 response options? What if someone reported that the child self-fed 75% of the time? Do they fall in the ‘all or most’ or 50/50 category and migrated to mixed? There is a need to be completed transparent about the specific questions that were asked regarding the adherence, the response options offered and how the investigators objectively and accurately categorized these responses into groups that meaningfully reflect adherence. Measuring adherence on a continuum or using more clearly defined categories to demonstrate adherence/non-adherence is necessary.

Minor issues

3. The authors need to carefully proof read the document for wording issues and punctuation. For example, there are sometimes full stops both before and after citations in brackets. Page 4 ‘both con consistencies’ which one can assume is meant to say ‘both consistencies’. Page 10, state ‘proposed’ when I think they mean reported? I would recommend changing the terminology from ‘subjected to an intervention’. ‘Subjected’ has different meanings in different contexts but can be inferred as meaning ‘to cause or force someone or something to experience something harmful, unpleasant’.

4. Intervention: It needs to clearer whether the intervention was individual or in groups. It can be inferred from page 5, they did it groups of 8 but this is not clear. In that section it reads like there may only be 8 pairs in total in each intervention arm, rather than 8 doing the intervention together in a group.

5. On Page 6 it states that parents could contact the nutritionist, did you measure how many parents did this and how often? It may be pertinent to consider if this was associated with adherence.

6. Within materials and methods there is repetition when describing the intervention, on page 4 labelled a/b/c/ then again on page 6 under intervention, please describe this only once.

7. Within data collection (page 7) it states ‘food frequency questionnaire and a food record prepared for this study’, please provide further details on the these measures; what specifically they included, how they were developed and their validity/reliability.

8. Page 7 – There are some wording issues; ‘The research team called all families to apply food records and ensure that they had no doubts, as well as to ensure the quality of the data.’ What do the authors mean by ‘applying the food records’ do they mean collecting the data? What is meant by ‘ensuring they have no doubts’, do the authors mean that the research team were available to respond to queries about the study or the feeding method should any arise? What did the researchers do if participants did ‘have doubts’? How many participants had 'doubts'? The nutritionists role if participants had 'doubts' needs to clearer as this is mentioned elsewhere in a different way.

9. Page 7 – Participants were ‘asked some questions regarding food consistency’, what were the questions? Please could the authors clarify if this was the two questions they state or were there more? There are issues with clarity in expression in this sentence that need to be addressed.

10. Page 7 – states ‘another researcher analysed the answers’, please state how they analysed them. Writing here needs to be more academic, the sentence states they compared the data with the data at 9 months, this could be removed as analysis is covered in the data analysis section, or the specific analyses techniques used to compare the data should be stated here.

11. Figure 1 a bit misleading with regard to ‘n’. I would recommend reporting the true n – the number of participants who completed the measures at each time-point ,(with the number missing at each time point noted), not the amount of people eligible for follow up.

12. Table 2 good be labelled more clearly. I would recommend stating ‘follows mode only’ and ‘follows consistency only’, this makes it clear you are referring to those who follow mode but not consistency, and consistency but not mode, and both.

13. Page 12 – some issues in interpretation and reporting of adherence findings, please report in a more meaningful way. Do the authors mean by 12 months 96 children were being fed with the mixed method? This does not mean they have ‘adhered’ to that method. It means some have not adhered to their randomised method and migrated to or chosen to follow mixed method. They cannot ‘adhere’ to something they were not allocate to.

14. The final paragraph of the discussion is weak and requires further development. There are also typo’s such as ‘..’.

6. PLOS authors have the option to publish the peer review history of their article (what does this mean?). If published, this will include your full peer review and any attached files.

Reviewer #1: No

Reviewer #2: No

---

## [Author Response · Author response to Decision Letter 0]

18 Jun 2023

Editor-in-Chief of Plos One,

 The authors are grateful for the appreciation and the opportunity to publish the manuscript entitled "Adherence to different complementary feeding methods in the first year of life: a randomized clinical trial". The authors also appreciate the time spent by the reviewers with important collaborations for the qualification of scientific production. Below are the responses to the comments.

 All changes in the manuscript are highlighted in yellow. A detailed point-to-point response follows for each reviewer is described below.

 The authors are available for questions or comments about the article.

Best regards,

The authors,

Universidade Federal do Rio Grande do Sul (UFRGS)

---

## [Decision Letter · Decision Letter 1]

25 Jul 2023

ADHERENCE TO DIFFERENT COMPLEMENTARY FEEDING METHODS IN THE FIRST YEAR OF LIFE: A RANDOMIZED CLINICAL TRIAL

PONE-D-22-20911R1

Dear Dr. Sanini Belin,

We’re pleased to inform you that your manuscript has been judged scientifically suitable for publication and will be formally accepted for publication once it meets all outstanding technical requirements.

Kind regards,

Erica Jane Cook, Ph.D

Academic Editor

PLOS ONE

Additional Editor Comments (optional):

Reviewers' comments:

Reviewer's Responses to Questions

**Comments to the Author**

1. If the authors have adequately addressed your comments raised in a previous round of review and you feel that this manuscript is now acceptable for publication, you may indicate that here to bypass the “Comments to the Author” section, enter your conflict of interest statement in the “Confidential to Editor” section, and submit your "Accept" recommendation.

Reviewer #2: All comments have been addressed

2. Is the manuscript technically sound, and do the data support the conclusions?

Reviewer #2: Yes

3. Has the statistical analysis been performed appropriately and rigorously? 

Reviewer #2: Yes

4. Have the authors made all data underlying the findings in their manuscript fully available?

Reviewer #2: Yes

5. Is the manuscript presented in an intelligible fashion and written in standard English?

Reviewer #2: Yes

6. Review Comments to the Author

Reviewer #2: (No Response)

7. PLOS authors have the option to publish the peer review history of their article (what does this mean?). If published, this will include your full peer review and any attached files.

Reviewer #2: No

---

## [Editor Report · Acceptance letter]

25 Oct 2023

PONE-D-22-20911R1 

Adherence to different complementary feeding methods in the first year of life: a randomized clinical trial 

Dear Dr. Sanini Belin:

I'm pleased to inform you that your manuscript has been deemed suitable for publication in PLOS ONE. Congratulations! Your manuscript is now with our production department. 

Kind regards, 

on behalf of

Dr. Erica Jane Cook 

Academic Editor

PLOS ONE